# Virus-infection in cochlear supporting cells induces audiosensory receptor hair cell death by TRAIL-induced necroptosis

Yushi Hayashi[1], Hidenori Suzuki[2], Wataru Nakajima[1], Ikuno Uehara[1], Atsuko Tanimura[1], Toshiki Himeda[3], Satoshi Koike[4], Tatsuya Katsuno[5], Shin-ichiro Kitajiri[5], Naoto Koyanagi[6], Yasushi Kawaguchi[6], Koji Onomoto[7], Hiroki Kato[8], Mitsutoshi Yoneyama[7], Takashi Fujita[8], Nobuyuki Tanaka[1]*

1 Department of Molecular Oncology, Institute for Advanced Medical Sciences, Nippon Medical School, Tokyo, Japan, 2 Division of Morphological and Biomolecular Research, Nippon Medical School, Tokyo, Japan, 3 Department of Microbiology, Kanazawa Medical University School of Medicine, Ishikawa, Japan, 4 Neurovirology Project, Tokyo Metropolitan Institute of Medical Science, Tokyo, Japan, 5 Department of Otolaryngology, Head and Neck Surgery, Kyoto University, Kyoto, Japan, 6 Division of Molecular Virology, Department of Microbiology and Immunology, The Institute of Medical Science, The University of Tokyo, Tokyo, Japan, 7 Division of Molecular Immunology, Medical Mycology Research Center, Chiba University, Chiba, Japan, 8 Laboratory of Molecular Genetics, Institute for Virus Research, Kyoto University, Kyoto, Japan

* nobuta@nms.ac.jp

**Data Availability Statement:** All relevant data are within the manuscript and its Supporting Information files. The cDNA microarray analysis data have been uploaded to the Gene Expression

## Abstract

Although sensorineural hearing loss (SHL) is relatively common, its cause has not been identified in most cases. Previous studies have suggested that viral infection is a major cause of SHL, especially sudden SHL, but the system that protects against pathogens in the inner ear, which is isolated by the blood-labyrinthine barrier, remains poorly understood. We recently showed that, as audiosensory receptor cells, cochlear hair cells (HCs) are protected by surrounding accessory supporting cells (SCs) and greater epithelial ridge (GER or Kölliker's organ) cells (GERCs) against viral infections. Here, we found that virus-infected SCs and GERCs induce HC death via production of the tumour necrosis factor-related apoptosis-inducing ligand (TRAIL). Notably, the HCs expressed the TRAIL death receptors (DR) DR4 and DR5, and virus-induced HC death was suppressed by TRAIL-neutralizing antibodies. TRAIL-induced HC death was not caused by apoptosis, and was inhibited by necroptosis inhibitors. Moreover, corticosteroids, the only effective drug for SHL, inhibited the virus-induced transformation of SCs and GERCs into macrophage-like cells and HC death, while macrophage depletion also inhibited virus-induced HC death. These results reveal a novel mechanism underlying virus-induced HC death in the cochlear sensory epithelium and suggest a possible target for preventing virus-induced SHL.

Omnibus (GEO) database (https://www.ncbi.nlm.nih.gov/geo/) under accession code GEO: GSE89556.

**Funding:** This work was also supported by Japan Society for the Promotion of Science (JP) (a Grant-in-Aid for Young Scientists (B) (15K20240) and a Grant-in-Aid for Research Activity start-up (25893299)) to YH. The funders had no role in study design, data collection and analysis, decision to publish, or preparation of the manuscript.

**Competing interests:** The authors have declared that no competing interests exist.

## Introduction

The World Health Organization (WHO) has reported that 360 million people, more than 5% of the world's population, suffer from disabling hearing loss [1]. Hearing loss is classified into two types; namely, conductive hearing loss and sensorineural hearing loss (SHL), the latter of which is the main type of hearing disability [2]. SHL is mainly caused by damage to cochlear hair cells (HCs), which function as audiosensory receptors [3]. Although the aetiology of SHL has not been identified, it has been suggested that viral infections such as cytomegalovirus (CMV), rubella, mumps, measles and herpes simplex virus (HSV) can cause it, especially with sudden SHL (SSHL), which usually develops in one ear within 72 h of infection [4,5]. Systemic corticosteroid administration is the primary treatment of choice for SSHL [6]. However, after corticosteroid treatment, hearing improvement is achieved in only 50% of patients, with 20% of them showing no change in hearing ability [7]. Because very little is known about the mechanisms underlying this disease and the anti-infection protection system in the inner ear, corticosteroids are still used despite their limited efficacies.

The inner ear is considered an immune-privileged site because the blood-labyrinthine barrier prevents access to it by the peripheral immune system [8]. The central nervous system, sensory organs, and gonads are separated from peripheral blood containing immune cells by a physical barrier to minimize any collateral tissue damage caused by an immune reaction [9]. It has been shown that lymphocytes reside only in the endolymphatic sac and are unlikely to participate in responses to pathogens [10], suggesting the innate immune system might be involved in the HC defence mechanism against pathogens. One of the earliest innate antiviral defence mechanisms is the type I interferon (IFN) system, and we have previously found that viral infection results in IFN-α/β production using the cochlear sensory epithelium isolated from newborn mice [11]. More recently, we have found that cochlear supporting cells (SCs), consisting of Hensen's cells, Claudius' cells, and greater epithelial ridge (GER or Kölliker's organ) cells (GERCs) in the neonatal immature inner ear, function as macrophage-like cells and protect adjacent HCs from pathogens [12].

Macrophages are found in all tissues and have roles in development, homeostasis, tissue repair and immunity, and disruption of their repair and homeostatic functions can cause many disease states including metabolic disease and cancer [13]. Moreover, blocking agents, such as monoclonal antibodies or cytokine antagonists of cytokine activity, especially those produced early in the inflammatory cascade such as tumour necrosis factor (TNF)-α and interleukin (IL)-6, are used for treating several inflammatory diseases, including rheumatoid arthritis and inflammatory bowel disease [14]. In the central nervous system, tissue resident macrophages (microglial cells) are considered to be a causative factor in brain diseases, including inflammatory diseases such as multiple sclerosis, degenerative diseases such as Alzheimer's disease (AD) or Parkinson's disease (PD), and psychiatric disorders such as depression or schizophrenia, and as such they are a potential target for the treatment of these diseases [15,16]. In relation to these findings, we found that cochlear SCs function similarly to tissue resident macrophages that protect HCs from pathogens [12]. Therefore, understanding the role(s) played by macrophages in infection and inflammation and their causative role(s) in disorder of their associated tissues is considered to be important for the successful treatment of various diseases.

In the present study, we investigated the effects of viral infection in the isolated murine newborn cochlear sensory epithelium, and found that virus-infected SCs and GERCs produced TNF-related apoptosis-inducing ligand (TRAIL), and TRAIL induced HC death by necroptosis. In addition, a necroptosis inhibitor efficiently suppressed the virus-induced HC death. These results suggest the mechanism underlying virus-induced SHL, and provide a potential target for treatment strategies against SHL.

## Materials and methods

### Experimental animals

Postnatal day 2 (sex: unknown) ICR mice (SLC), *interleukin 6* (*Il6*) null mice (Jackson Laboratory) and *interferon (alpha and beta) receptor 1* (*Ifnar1*) null mice (B&K Universal) were used in this study. The animal experiment protocol was approved by the Ethics Committee on Animal Experiments of Nippon Medical School. It was carried out in accordance with the guidelines for Animal Experiments of Nippon Medical School and the guidelines of The Law and Notification of the Government of Japan, as well as the ARRIVE guidelines. Mice were maintained 12 hour light/12 hour dark cycle at 20–24˚C with 40–70% humidity. They were allowed to have free access to standard laboratory mouse chow, MF (Oriental Yeast Co., ltd. Tokyo, Japan), and free access to drinking water. They were housed at a maximum number of five. All mice were checked for stress each day. Mice were euthanized by cervical dislocation for further experiments.

### Preparation and treatment of cochlear sensory epithelium explant cultures

Cochlear sensory epithelia were resected and cultured as previously described [11]. All experiments began after an overnight incubation of each cochlea with 300 μl medium [Dulbecco's modified Eagle's medium (DMEM; Sigma-Aldrich Inc., St. Louis, MO, USA) supplemented with D-glucose (6 g/l) and penicillin G (150 μg/ml)] for explant cultures. Each cochlea was then transferred to 400 μl medium containing $3.0 \times 10^7$ pfu/ml TMEV, which is an RNA virus. TMEV (GDVII strain) was propagated from viral cDNA and BHK21 cells [17]. Cultures were maintained for 9–12 h until TMEV began to infect the SCs. At 16–21 h, TMEV began to infect the GERCs and HC death was observed. HC death was almost completed by 24 h. To analyze the paracrine influence of SC/GERC-secreted tumor necrosis factor-related apoptosis-inducing ligand (TRAIL), the TMEV-infected cochleae were incubated in rat anti-TRAIL antibody (0.01 mg/ml, GeneTex, clone N2B2) for 24 h. Phosphorylated mixed lineage kinase domain-like (p-Mlkl) expression in HCs suggested necroptosis, therefore necroptosis inhibitors such as necrostatin-1 (Abcam) or ponatinib (Selleckchem) were used at the specified concentrations for up to 24 h. Steroids such as dexamethasone are primarily used to treat SSHL in the clinic, although its effectiveness is limited and the mechanisms of action remain poorly understood. To compare the effectiveness of steroids with necrostatin-1 and to determine the unidentified mechanisms of steroids, 1.0 μM dexamethasone (Wako) was added to medium containing TMEV and cultivated for up to 24 h. Clophosome™ (91 μg/ml for up to 24 h; Funakoshi), which is a liposome-clodronate reagent, was used to deplete activated SCs and GERCs as macrophages to determine whether these macrophage-like cells injured HCs.

### Immunohistochemistry

For whole mount immunohistochemistry, samples were fixed at room temperature (RT) for 15 min in 4% paraformaldehyde in 0.1 M phosphate buffer (pH 7.4) and then rinsed with PBS. All specimens were incubated in blocking solution at RT for 30 min in 10% goat serum with 0.2% Triton X-100 for all antibodies except the anti-cleaved caspase 3 antibody or for 15 min in 0.2% Triton X-100 and 15 min in 1% BSA in 0.2% Triton X-100 when colabelled with cleaved caspase 3. The primary antibodies used in this study were as follows: rabbit polyclonal anti-Myosin VIIA (Myo7a) (25–6790; 1:1000, Proteus Biosciences), mouse monoclonal anti-double-stranded RNA (dsRNA) (J2; 1:800, K1; 1:2000, English & Scientific Consulting), rabbit polyclonal anti-cleaved caspase 3 (#9664; 1:100, Cell Signaling), rabbit polyclonal anti-DR4 (GTX28414; 1:1000, GeneTex), rabbit polyclonal anti-DR5 (ab8416; 5 μg/ml, Abcam), and

rabbit monoclonal anti-p-Mlkl (phospho S345) [EPR9515(2); 1:1000, Abcam]. Actin filaments were visualized with Alexa 594- or 633-labelled phalloidin (1:100, Invitrogen). The primary antibodies were visualised with Alexa 488- or 546-conjugated anti-rabbit or anti-mouse IgGs (1:1000, Invitrogen). Samples with nuclear staining were incubated in PBS containing 1 μg/ml DAPI (Invitrogen). Fluorescence images were captured under an FV1200 confocal microscope (Olympus). Computational section images were reconstructed by FV10-ASW (Olympus) after capturing images every 0.2–0.5 μm. Whole mount images of HCs were obtained by superposing images from the bottom to the top of HCs after capturing images every 0.5 μm. Whole mount images of virally infected, activated SCs and GERCs were obtained from single slices or several slices overlapped at the SC or GERC level after capturing images every 0.5 μm.

## Cell counts

The numbers of inner and outer HCs in the sensory epithelium were counted along a 100-μm longitudinal distance in the basal- to mid-turn of each explant. As described above, we estimated from the basal- to mid-turn of the cochlea, because Hensen's cells and Claudius' cells in the apical turn overlapped with the inner side of the basal-turn. HCs with remaining stereocilia were counted as live HCs, and stereocilia were identified with phalloidin, which detects actin filaments.

## Electron microscopy

Electron microscopy analysis was performed as described previously [18,19], with slight modification. Mouse inner ears were observed by transmission electron microscopy (TEM). The specimens were fixed with 2% glutaraldehyde in 0.1 M phosphate buffer (pH 7.4) for 60 min, washed five times in 0.1 M phosphate buffer, and post-fixed with 1% osmium tetroxide for 60 min at 4˚C. The fixed inner ears were dehydrated with a graded ethanol series, and embedded in EponA2 according to the conventional method. Thin sections were cut with a diamond knife, stained with uranyl acetate and lead citrate, and examined with a JEM-1010 transmission electron microscope (JEOL) at an accelerating voltage of 80 kV.

## RNA extraction, qRT-PCR, and microarray

For qRT-PCR, an RNeasy Micro Kit (Qiagen) was used to extract total RNA from three cochlear sensory epithelia that were cultivated under the following conditions for 16 h: DMEM alone; DMEM with TMEV; DMEM with TMEV and dexamethasone; DMEM with TMEV and Clophosome™; DMEM with LPS. cDNA was synthesized from DNase-treated total RNA using a PrimeScript RT reagent Kit (Takara Bio). Synthesized cDNA was subsequently mixed with TaqMan Universal PCR Master Mix (Applied Biosystems) in the presence of commercial TaqMan primer-probe sets of interest (Applied Biosystems). Real-time PCR quantification was performed using the ABI StepOnePlus Real-Time PCR System (Applied Biosystems). All reactions were performed in triplicate. Relative mRNA levels were calculated using the ΔΔCt method. For the invariant control, we used *actin beta* (*Actb*). For microarray, total RNA was extracted from nine mock explants, nine explants treated with 1000 ng/ml LPS for 9 or 16 h, and 14 explants infected with TMEV for 9 or 16 h using the RNeasy Micro Kit. Filgen (a biological technical service company) performed the microarray analysis using a GeneChip Mouse Gene 2.0 ST Array (Affymetrix). These data have been uploaded to the Gene Expression Omnibus (GEO) database (https://www.ncbi.nlm.nih.gov/geo/) under accession code GEO: GSE89556.

## Statistical analysis

The data are expressed as the mean ± standard error. One-way ANOVA was used for the HC counts in necrostatin-1 and ponatinib assays or *Ifnar1* KO, *Il6* KO, and LPS assays. *P*-values < 0.05 were considered statistically significant. Post hoc tests were performed using the Bonferroni method. *P*-values < 0.005 (necrostatin-1 assay) 0.0167 (ponatinib assay) or 0.0033 (*Ifnar1* KO, *Il6* KO, and LPS assays) were considered statistically significant. In the other experiments for which HC counts were needed, statistical analyses were performed by unpaired *t*-tests, in which *P*-values < 0.05 were considered statistically significant. In all qRT-PCR analyses, unpaired *t*-tests were used, in which *P*-values < 0.05 were considered statistically significant.

## Results

### Virus-infected SCs and GERCs induce HC death

We previously found that Theiler's murine encephalomyelitis virus (TMEV) infection in the isolated murine newborn cochlear sensory epithelium induces IFN-α/β production [11]. TMEV is a small RNA picornavirus commonly used as an experimental model system for blood-brain barrier disruption [20]. Recently, we observed that TMEV infection is mainly established in SCs and HC infection is rarely observed in the presence of IFN-α/β produced by SCs that function as macrophage-like cells [12]. We also observed that SC infection is established in the early stage of TMEV infection (9 h after virus infection) and GERC infection is established in the later stage (16 h after virus infection) [12]. To understand the influence of virus infection in SCs on HCs, we analyzed the cell status of HCs using the same experimental system. It has been reported that TMEV infects macrophages [21]. Indeed, we have observed that TMEV infected almost all SCs and GERCs in our experimental system [12]. Therefore, although TMEV is not a virus that causes SHL in humans such as CMV, we used this experimental system to investigate the effects of infection with cochlear SCs that are protective cells against virus infection in mice. As a result, we found that HC death occurred at 16 h after a virus infection, despite very few virus-infected HCs being present (Fig 1A). Most HCs died within 24 h of infection (Fig 1B). Next, we analysed the effects of virus-inducible cytokines IFN-α/β [22], which induce cell death in virus-infected cells, and IL6 that has been reported to regulate apoptosis of TMEV-infected cells [23]. As shown in Fig 1C–1E, TMEV-induced HC death was not affected in IFN-α/β receptor 1 (*Ifnar1*)$^{-/-}$ and IL6 (*Il6*)$^{-/-}$ sensory epithelium. Moreover, signals of Toll-like receptors (TLRs), which recognize microbial components, induce apoptosis [24], but lipopolysaccharide (LPS) that activates TLR4 did not induce HC death (Fig 1C–1E). Although the time of infection establishment was different between SCs and GERCs, there were no significant differences between loss of inner HCs (IHCs) and outer HCs (OHCs) (Fig 1E). These results suggest that signal(s) other than virus components and virus-inducible cytokines IFN-α/β and IL6 induce HC death.

To clarify this death signal, we performed cDNA microarray analysis and found that expression of the cell death-inducing ligand *Trail* was induced by the viral infection, but not LPS (Fig 1G). TRAIL, a TNF superfamily protein, mediates killing of virus-infected cells and is involved in the pathogenesis of multiple virus-induced disorders [26]. It has also been shown that virus infection and IFN-α/β stimulation of immune cells induce expression of TRAIL [27].

### TRAIL produced by virus infection induces HC death

TRAIL, a potent stimulator of apoptosis, works by binding to DR4 (also known as TRAILR1) and DR5 (also known as TRAILR2) death receptors [26,28]. Expression of DR4 and DR5 was

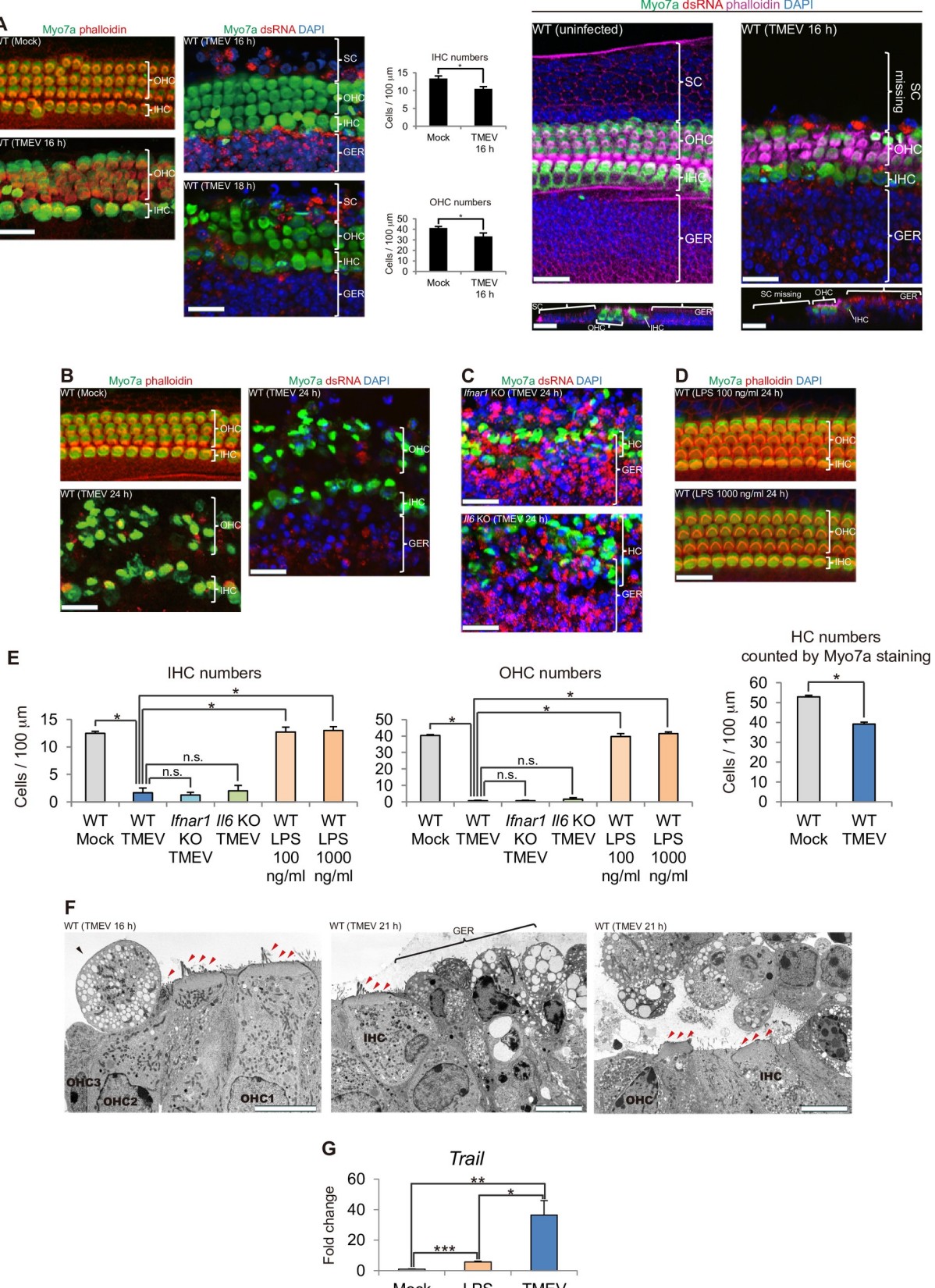

**Fig 1. Temporal analysis of HC death following viral infection.** (**A**) At 16 h after TMEV infection of the cochlear sensory epithelium, HC death was initiated in spite of the presence of very few virus-infected HCs [*$P < 0.05$, $t$-test, Mock: n = 5, TMEV: n = 4, outer HCs (OHCs), inner HCs (IHCs)]. Even at 18 h after TMEV infection when HC death was progressing, very few dsRNA-positive HCs were observed. At this time point (16 h after TMEV infection), the majority of SCs were lost by TMEV infection, which was confirmed by the computational section (see the uninfected sample). (**B**) Severe HC damage was observed at 24 h after TMEV infection with migration of dsRNA-positive SCs and GERCs. Most dying HCs were still negative for dsRNA. Migration of virus-infected SCs and GERCs has been reported previously [12]. (**C**) *Ifnar1* KO and *Il6* KO mice experienced SC and GERC migration to the HC layer with severe HC damage during the virus infection. (**D**) LPS treatment did not induce HC death with no migration of SCs and GERCs. (**E**) IHC and OHC numbers (counted by phalloidin staining) at 24 h of incubation with TMEV or LPS (IHC and OHC: $P < 0.0001$, ANOVA; *$P < 0.0001$, Bonferroni). Both IHCs and OHCs were reduced significantly in TMEV-infected WT cochleae (n = 3) compared with WT with mock treatment (n = 6). Even when Ifnar1 (n = 4) or Il6 (n = 3) was deficient in cochlear sensory epithelia, HC death occurred with TMEV infection similarly to that seen in WT mice, which suggested that these cytotoxic cytokines do not induce HC death. Almost all HCs survived when treated with LPS (100 ng/ml: n = 4, 1000 ng/ml: n = 4). HC death during TMEV infection was also confirmed by HC counting based on Myo7a staining (*$P < 0.0001$, $t$-test, WT mock: n = 6, WT TMEV: n = 3). These observations indicate that some kinds of secreted proteins, which were secreted when infected with TMEV, except for IFNs and IL6, but not secreted when incubated with LPS, are the cause of HC death. (**F**) HC stereocilia (red arrowheads) degenerated over time after activation of SCs (black arrowhead) and GERCs as macrophages (TEM; OHC1: First row of OHCs, OHC2: Second row of OHCs, OHC3: Third row of OHCs). Cytoplasmic vacuolisation has been found in many virus-infected cells [25]. (**G**) qRT-PCR analysis revealed that virus infection (n = 5) was more effective to upregulate *Trail* expression compared with LPS treatment (n = 3) or mock treatment (n = 6) at 16 h (*$P < 0.05$, **$P < 0.01$, ***$P < 0.0001$, $t$-test). Scale bars: Immunostaining, 20 μm; TEM, 10 μm. Error bars, standard errors.

found in HCs, but rarely in SCs (Fig 2A). We previously found that TMEV-infected SCs and GERCs express macrophage marker proteins and perform phagocytosis, which indicate that SCs and GERCs are macrophage-like cells [12]. It has been shown that TRAIL is induced in virus-infected macrophages [27], and that *Trail* is a transcriptional target of virus-induced transcription factor interferon regulatory factor 3 [29]. Indeed, in SHIELD (Shared Harvard Inner-ear Laboratory Database [30]), macrophage marker *F4/80* and SC marker *Sox2* were expressed in SC fractions [GFP(-)] and HC markers *Myo7a*, *prestin*, and *Pou4f3* were expressed in the HC fraction [GFP(+); S1 Fig]. Moreover, *Trail* was expressed in SC fractions that were higher than HC fractions [especially at embryonic day 16 and postnatal day 0; S1 Fig]. Additionally, TMEV infections were mainly established in SCs and infections of HCs were rarely observed (Fig 1A). These findings suggest that TRAIL was produced by SCs, which function as macrophages, after TMEV infection. However, it cannot be ruled out that factor(s) produced by virus-infected SCs act on HCs to induce TRAIL. Moreover, The SC fraction is a cell population other than the HC fraction of sensory epithelial cells, which was thought to be absent of immune cells, but it cannot be completely ruled out that this population contains small numbers of macrophages and lymphocytes. However, these findings suggest that TRAIL was produced by SCs that function as macrophages after TEMV infection.

As shown in Fig 2B, HC death was efficiently suppressed by a TRAIL-neutralizing antibody. The specificity and neutralizing activity of the used antibody against TRAIL (monoclonal N2B2 antibody) have been shown previously [31,32]. Although it is possible that loss of SCs leads to HC death, loss of IHCs and OHCs was effectively blocked in the presence of the TRAIL-neutralizing antibody in spite of SC loss. In this result, the decrease of IHCs was almost suppressed by the TRAIL-neutralizing antibody, but a decrease of OHCs was slightly observed (Fig 2B). Regarding this difference, it is possible that there was a difference in the local TRAIL concentration and the effect of SC loss, but this has not been clarified at this time. These results indicated that TRAIL was the effector for virus-infected SC- and GERC-induced HC death. However, because four TRAIL receptors, which include DR4 and DR5, have been identified in humans [33], it is not to be elucidated whether only the binding of DR4 and DR5 expressed in HCs is important for HC death by TRAIL. However, recombinant TRAIL protein itself was not sufficient to induce HC death (Fig 2C). Therefore, these results indicated that HC death was triggered by TRAIL and suggested that other factor(s) are required to induce full death. Considering human diseases, it has been shown that the loss of both HCs and SCs occurs in many human SSHL cases [34], which suggests that virus-infected SCs induce HC death in SSHL.

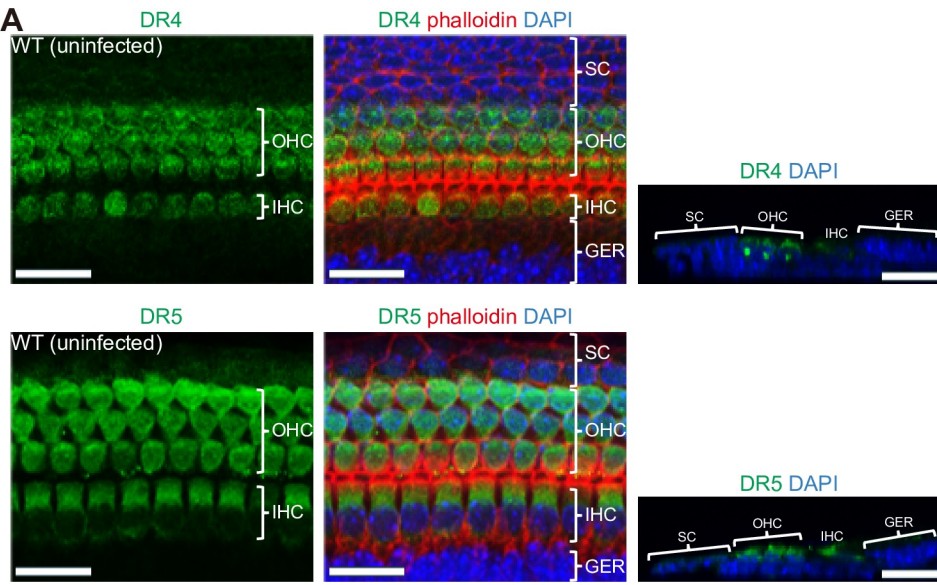

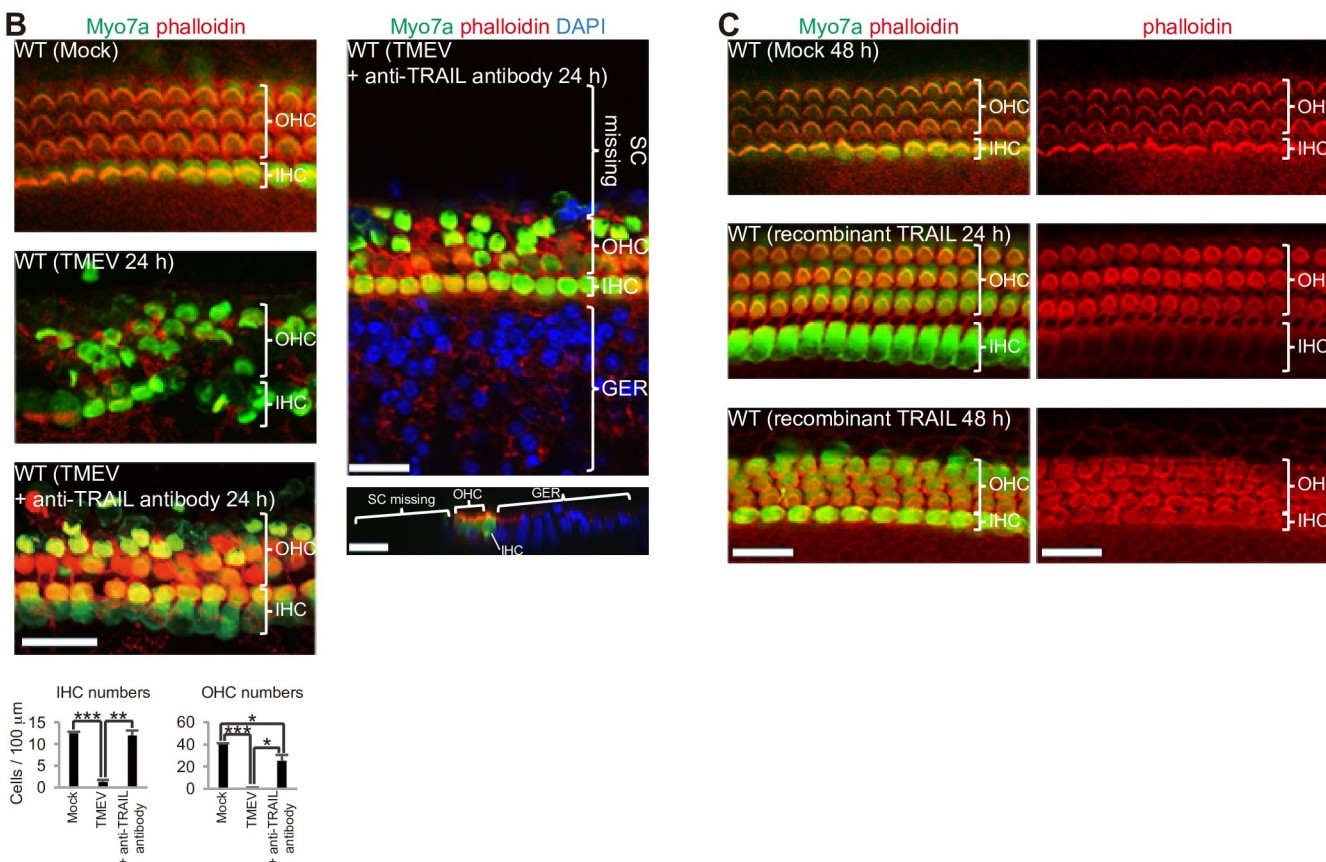

**Fig 2. TMEV infection of SCs and GERCs results in TRAIL-mediated HC death.** (**A**) HC expression of TRAIL receptors DR4 (green) and DR5 (green). (**B**) TRAIL-neutralizing antibody (5 μg/ml) attenuated HC damage (\**P* < 0.01, \*\**P* < 0.001, \*\*\**P* < 0.0001, *t*-test, Mock: n = 6, TMEV: n = 4, + anti-TRAIL antibody: n = 3). While HCs were protected by anti-TRAIL antibody during TMEV infection, the death of SCs was not attenuated, indicating that the anti-TRAIL antibody has its effect directly on HCs. (**C**) Stereocilia of HCs was almost intact when treated with recombinant TRAIL protein (6 μg/ml) for 24 h, but long term exposure to recombinant TRAIL protein (48 h) disorganised and deformed the stereocilia. Scale bars, 20 μm. Error bars, standard errors.

## Virus-induced HC death is not caused by apoptosis

Apoptosis is the most commonly observed cell death during a viral infection, and it is considered that host cells eliminate virus-infected cells via apoptosis, which aborts further viral infection [35]. During SC- and GERC-induced HC death, the activation of caspase 3, a mediator of apoptosis, was observed in the SCs and GERCs, but not in the HCs (Fig 3A). This suggests that the viral infection induced apoptosis in the virus-infected SCs, but that SC-mediated HC death is not caused by apoptosis. It has been shown that the effect of aminoglycoside, a drug known to be ototoxic to HCs, can be suppressed by a caspase inhibitor [36], indicating that this type of cell death is caused by apoptosis. However, the mechanism underlying this phenomenon is not fully understood [37]. Therefore, we next analysed the expression of inflammation and macrophage markers in the presence of aminoglycoside; however, the expression of these markers in SCs was not induced by gentamicin, an aminoglycoside antibiotic (Fig 3B). These results suggest that the virus-induced HC death mechanism differs from that induced by ototoxic drugs.

## Virus-induced death of HCs is mediated by TRAIL-induced necroptosis

Programmed cell death plays a fundamental role in animal development and tissue homeostasis, and is regulated by a variety of mechanisms such as apoptosis, necroptosis, ferroptosis and pyroptosis, among others [39]. Among these cell deaths, we found that addition of necrostatin-1 [40] and ponatinib [41] necroptosis inhibitors efficiently supressed SC- and GERC-induced HC death (Fig 4A and 4B). Indeed, in addition to expression of apoptosis-related genes, expression of necroptosis-related genes was induced in TMEV-infected cochlear sensory epithelium (S2 Fig). Moreover, although necrostatin-1 effectively suppressed HC loss, there was no suppression of SC loss (Fig 4A). This result suggests that the necroptosis inhibitor affected HC death, but did not inhibit SC death, which was caused by apoptosis (Fig 3A). Moreover, it has been suggested that reactive oxygen species (ROS) induce HC death [42]. However, genes involved in ROS production, such as *Nox1-4*, *Tp53*, *Ptgs2* (prostaglandin-endoperoxide synthase 2), and *Tnfa* (tumour necrosis factor-α), showed no marked induction similar to apoptosis- and necroptosis-related genes by virus infection. Furthermore, the SC- and GERC-induced phosphorylation of the necroptosis-regulator (mixed lineage kinase domain-like, Mlkl) [43], which was also observed in the HCs, was blocked by necrostatin-1 treatment (Fig 4C). Moreover, it has been shown that TRAIL induces necroptosis under some conditions [44]. Therefore, these results suggest that SC- and GERC-induced HC death was caused by necroptosis [40]. We quantified HC death by the number of HCs along a longitudinal distance of 100 μm. However, TMEV treatment induced SC death, which resulted in disorganization of cochlear tissues. Therefore, it is possible that virus-induced disorganization led to the appearance of lower numbers of hair cells when counted in the same width. However, as shown in Fig 4A, necrostatin treatment, especially 400 mm treatment, significantly inhibited loss of HC numbers (HC number was not markedly reduced compared with that observed without virus infection shown in Fig 2B), whereas most SCs were lost. To more accurately assess HC death, we believe that the effects of disorganization in cochlear tissues need further analysis.

## Prevention of viral infection-induced HC death by targeting the macrophage functions of SCs and GERCs

Currently, corticosteroids are the primary effective therapeutic agents for SSHL [45]. To understand the role played by corticosteroids in virus infection-induced HC death, we

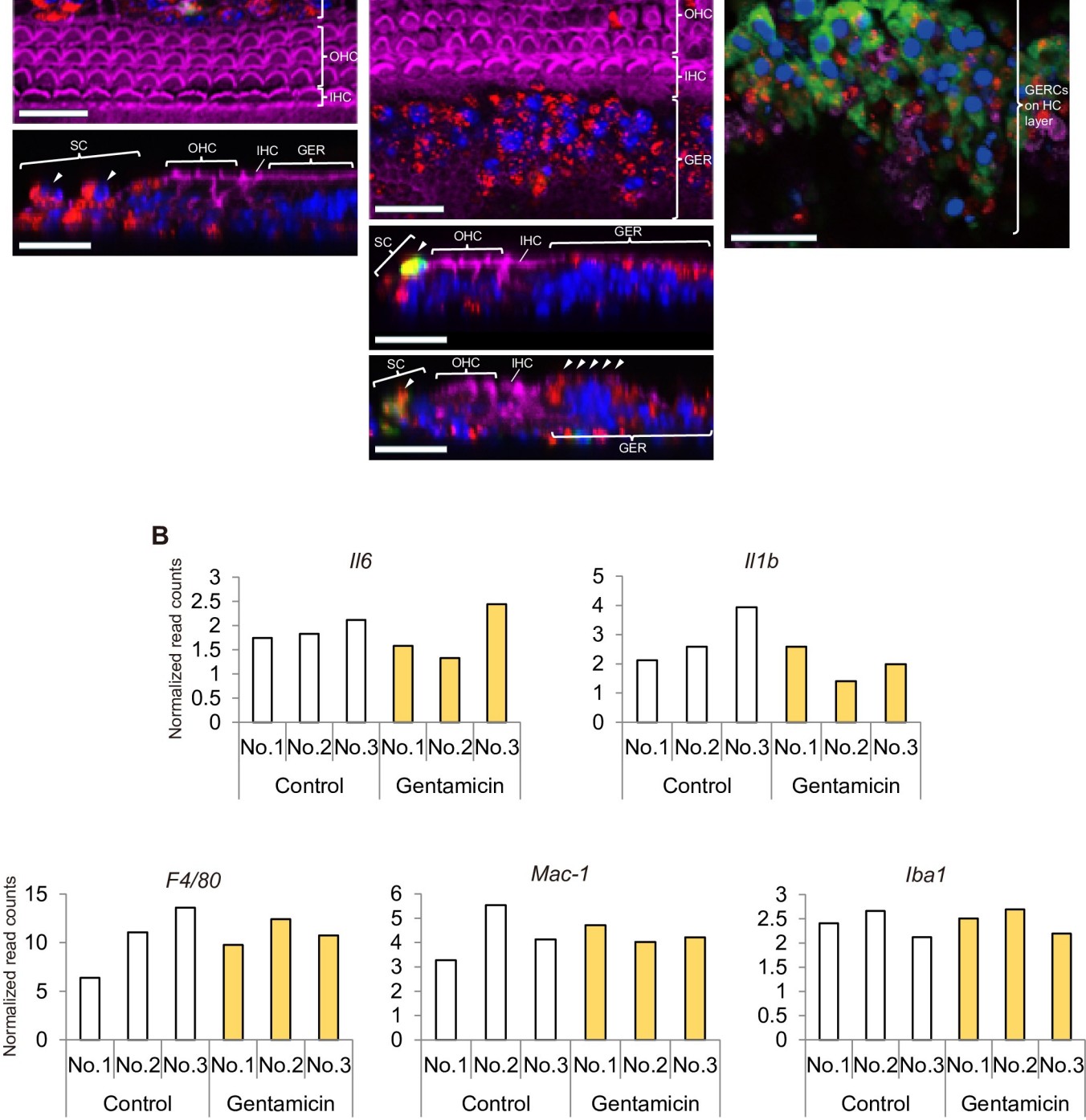

**Fig 3. Virus-infected SC- and GERC-induced HC death is not mediated by apoptosis.** (**A**) Cleaved caspase 3 (green) was expressed in TMEV-infected SCs and GERCs (arrowheads in sections), but not in HCs, which indicated that apoptosis did not occur in HCs. At 20 h after infection (right panel), most SCs had died by apoptosis and infected GERCs had migrated onto the HC layer, as described previously [12]. (**B**) Tao et al. deposited RNA sequence datasets for sorted HCs (GFP positive) and surrounding cells including SCs (GFP-negative non-HCs) from explant cultures of an Atoh1-GFP organ of Corti treated with gentamicin for 3 h in the NCBI GEO database (GSE66775) [38]. In this database, we focused on macrophage- and inflammation-correlated gene expression changes in a non-HC population including SCs to compare virus infection with aminoglycoside-related injury. Here, we extracted inflammation markers *Il6* and

*Il1b*, and macrophage markers *F4/80*, *Mac-1*, and *Iba1*. No significant difference was observed in the expression levels of these genes in the non-HC population, which included SCs, between the control (n = 3) and gentamicin-treated group (n = 3). Scale bars, 20 μm.

investigated whether dexamethasone administration would result in more numbers of HCs in TMEV infections. In fact, dexamethasone suppressed virus infection-induced HC death (Fig 5A). Macrophage marker expression was also inhibited in the SCs and GERCs (Fig 5B), suggesting that corticosteroids inhibit virus-induced HC death by inhibiting virus-induced

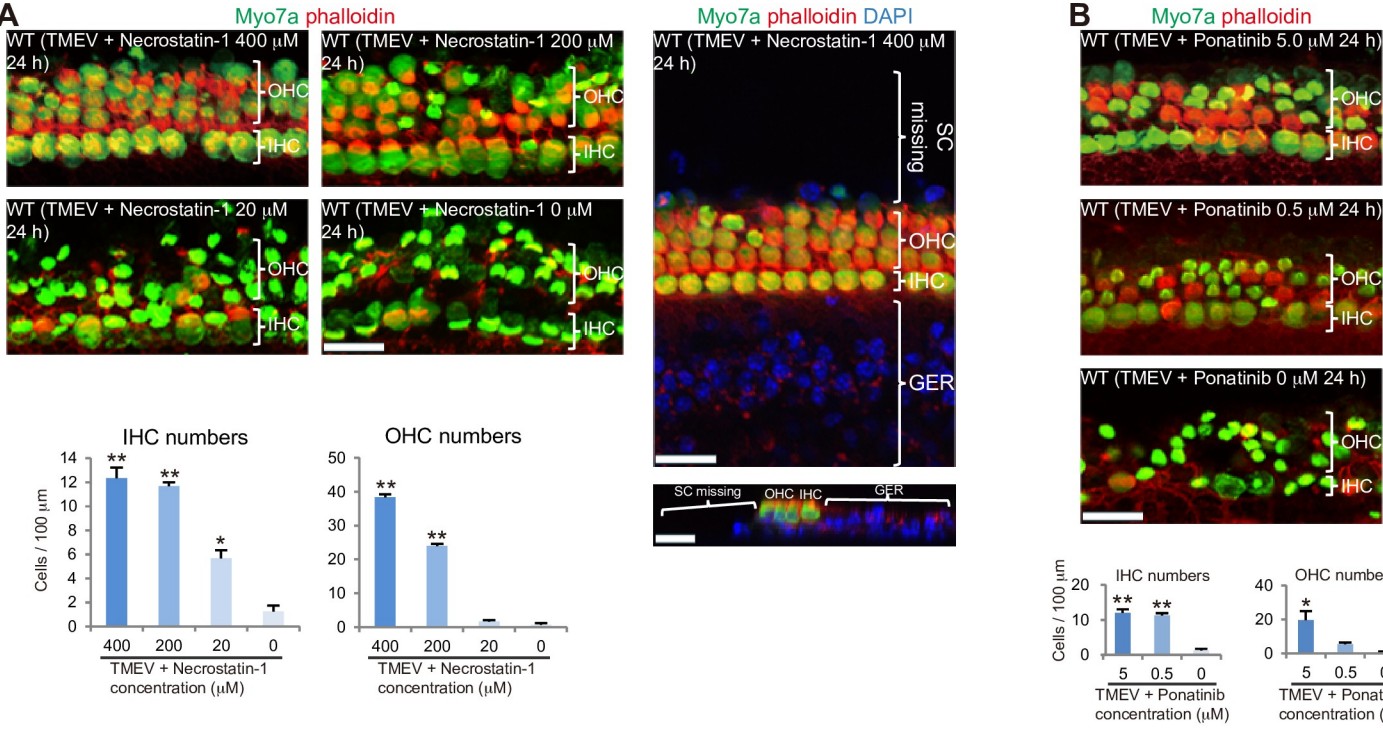

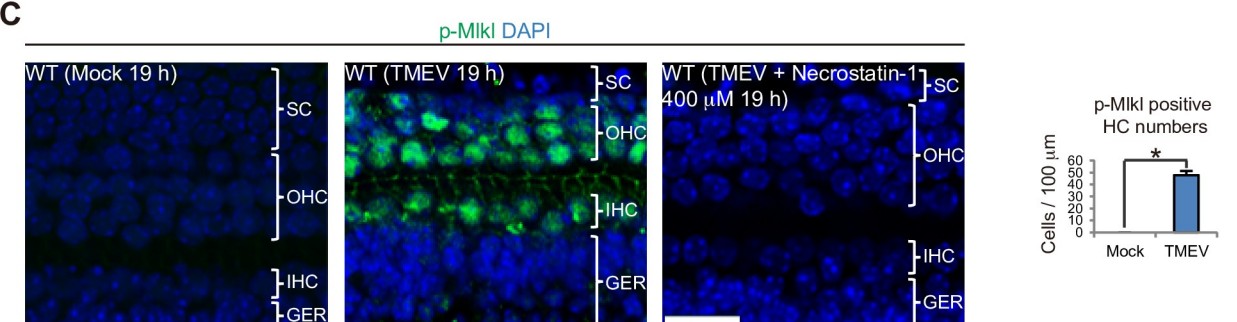

**Fig 4. Necroptosis inhibitors supress HC damage.** (**A, B**) Necroptosis inhibitors necrostatin-1 (**A**) (IHC and OHC: $P < 0.0001$, ANOVA, 400 μM: n = 3, 200 μM: n = 3, 20 μM: n = 3, 0 μM: n = 4) and ponatinib (**B**) (IHC: $P < 0.0001$, OHC: P = 0.0021, ANOVA, 5 μM: n = 3, 0.5 μM: n = 4, 0 μM: n = 4) both attenuated HC death (*$P < 0.001$, **$P < 0.0001$, Bonferroni). While HCs were protected by Necrostatin-1 during TMEV infection, the death of SCs was not diminished, demonstrating that Necrostatin-1 functions directly to HCs as well as the TRAIL-neutralizing antibody. (**C**) p-Mlkl (green) expression in HCs induced by TMEV infection was inhibited by necrostatin-1. Thus, SCs and GERCs both induced HC necroptosis via the TRAIL-death receptor-signalling cascade. *$P < 0.001$, *t*-test, mock: n = 3, TMEV: n = 3. Scale bars, 20 μm. Error bars, standard errors.

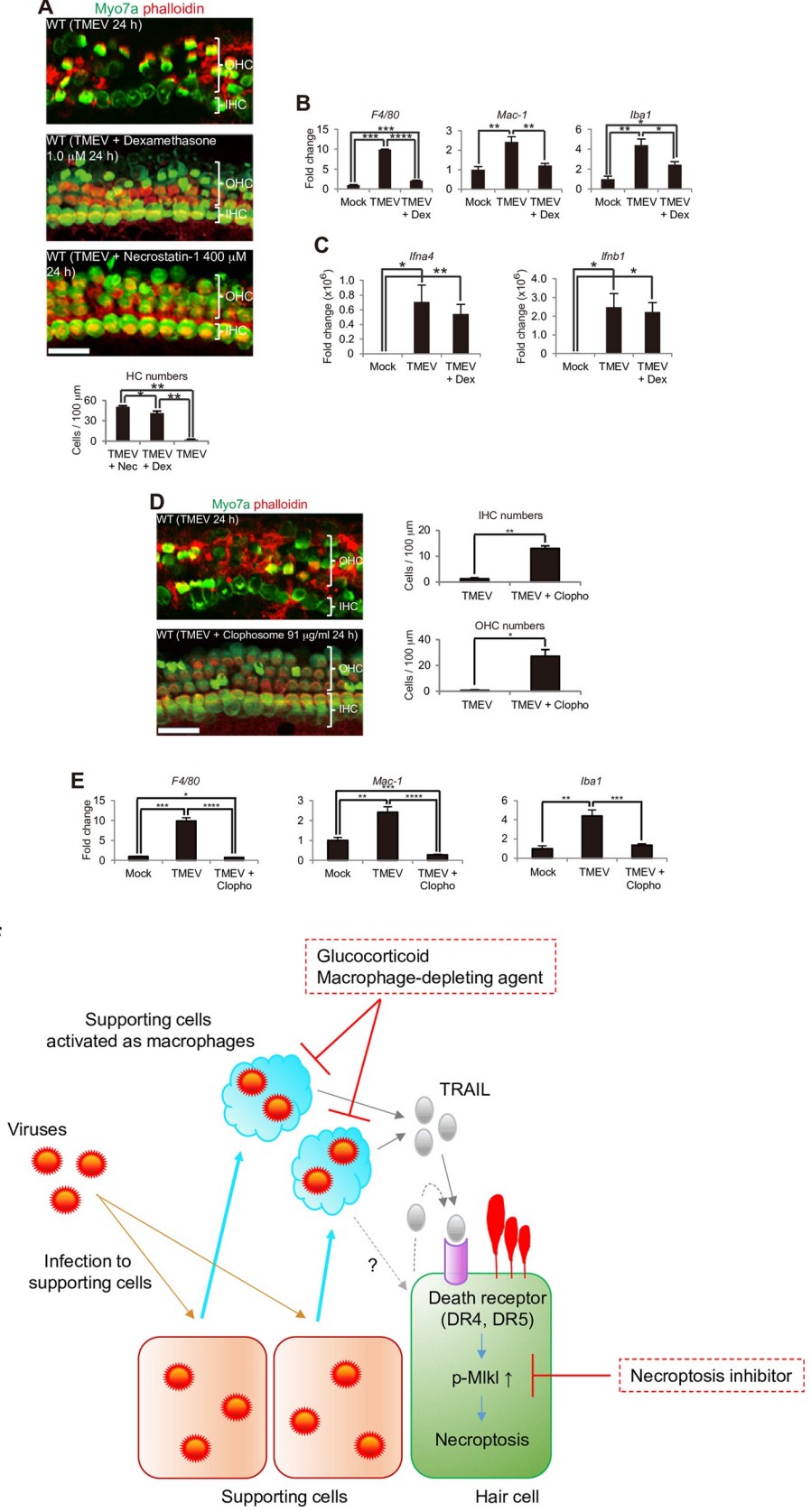

**Fig 5. Targeting the macrophage functions of SCs and GERCs supresses HC damage.** (**A–C**) Corticosteroid dexamethasone (Dex) inhibited HC damage. (**A**) (*$P < 0.05$, **$P < 0.0001$, $t$-test; TMEV + Nec (necrostatin-1): n = 3, TMEV + Dex: n = 3, TMEV: n = 4). Dex downregulated macrophage marker expression (**B**) (Mock: n = 3, TMEV: n = 4, TMEV + Dex: n = 5), but did not downregulate type I IFN expression at 16 h (**C**) (Mock: n = 5, TMEV: n = 5, TMEV + Dex: n = 5) (*$P < 0.05$, **$P < 0.01$, ***$P < 0.001$, ****$P < 0.0001$, $t$-test). The mechanism underlying the ability of Dex to treat SHL involves downregulation of SC and GERC macrophage functions in spite of type I IFN expression not being suppressed, which subsequently protects HCs from viral infection. (**D, E**) SC depletion following Clopho treatment leads to HC survival during virus infection. Clopho administration removed activated SCs as macrophages, thereby attenuating HC damage during TMEV infection (**D**) (*$P < 0.01$, **$P < 0.0001$, $t$-test, TMEV: n = 4, TMEV + Clopho: n = 3). Clopho administration to TMEV-infected explants downregulated macrophage marker expression (**E**) (16 h; *$P < 0.05$, **$P < 0.01$, ***$P < 0.001$, ****$P < 0.0001$, $t$-test, Mock: n = 3, TMEV: n = 4, TMEV + Clopho: n = 6). (**F**) A diagram indicating the mechanism of HC death by SCs activated as macrophage-like cells and HC protection by the glucocorticoid, macrophage-depleting agent, and necroptosis inhibitor. Scale bars = 20 μm. Error bars, standard errors.

macrophage changes in SCs and GERCs. Conversely, IFN-α/β expression was unaffected, indicating the therapeutic benefit of corticosteroids in virus-induced SHL without suppressing the antiviral effect of IFN-α/β (Fig 5C). The same effect was also observed with Clophosome™ (Clopho), which contains macrophage-depleting anionic lipids [46] (Fig 5D and 5E). Therefore, these results suggest that in addition to necroptosis, macrophages are a candidate target for the prevention of virus-induced SSHL.

## Discussion

We recently investigated the defence mechanism used by audiosensory receptor HCs against viral and bacterial infections, and found that the SCs and GERCs surrounding the HCs function like macrophages and protect these cells [12]. In the present study, we found that when excess virus infected the SCs and GERCs, the SCs changed from "guardians" to "aggressors" and caused HC death. It has been widely shown that inflammatory macrophages are initially beneficial to the body because they facilitate the clearance of invading organisms; however, they also trigger substantial collateral tissue damage, resulting in various diseases [13]. At present, we still do not know whether the same phenomena also occur in adult humans in response to a viral infection. Because it has been reported that both HCs and SCs are absent in many human SSHL cases [34], it is conceivable that in response to a viral infection virus-induced apoptosis of SCs and TRAIL-induced necroptosis of HCs concurrently occur in the inner ear, a process that may occur in SSHL. It has been shown that macrophages play roles in development, homeostasis, tissue repair and host defences against infection, and also play pathophysiological roles that result in chronic inflammation [13]. Furthermore, it is now well appreciated that many major diseases and conditions (e.g., atherosclerosis, obesity, diabetes and cancer) are associated with chronic inflammation, and that macrophages play a role in them [13,47]. Therefore, macrophage-targeting therapies are in development against such diseases [48,49].

TRAIL, a TNF superfamily member protein, induces apoptosis through the caspase activation pathway [33]. This protein is also known to induce necroptosis under certain conditions, such as acidic pH, depletion of the cellular inhibitor of apoptosis (cIAP) or TNF receptor-associated factor 2 (TRAF2), via the receptor-interacting serine/threonine protein kinase (RIPK) 1 and RIPK3 [33]. It has been shown that necroptosis promotes further cell death and neuroinflammation during the pathogenic processes of several neurodegenerative diseases, including multiple sclerosis, amyotrophic lateral sclerosis (ALS), AD and PD, through the death receptors of TNF superfamily members [50]. In living slices of human brain tissue, TRAIL was found to induce cell death in neurons and glias, suggesting that TRAIL acts as a destructive effector molecule in the human brain [51]. Furthermore, TRAIL is expressed in the brains of patients with AD and is completely absent from the brains of patients without AD [52]. In

human multiple sclerosis lesions, and in mouse brains after the induction of experimental autoimmune encephalomyelitis, TRAIL is upregulated, predominantly by activated microglia and invading immune cells [53]. This suggests that TRAIL contributes to pathogenesis in neurological disorders. Our present results show that, as well as in the central nervous system, the TRAIL produced by activated macrophage-like cells is involved in sensory receptor disorders. Additionally, necrostatin-1, a necroptosis inhibitor, has shown efficacy in improving tissue injuries in animal models of diseases ranging from ischemic brain, kidney and heart injuries, to multiple sclerosis, ALS, and AD [54]. However, although several necroptosis inhibitors are effective against inflammation-mediated disorders [40,41], at present, only a few have passed to the clinical testing stage [55]. Further development of clinically useful necrosis inhibitors for the treatment of such diseases is expected.

Although it has been observed rarely, SSHL may occur in patients with COVID-19 (coronavirus disease 2019) [56]. In relation to this, it has been reported that respiratory tract infection by coronavirus, especially SARS (severe acute respiratory syndrome)-coronavirus, causes marked elevation of TRAIL production [57]. At present, corticosteroids are the primary effective therapeutic agents for SSHL [45]. However, the use of corticosteroids for treatment of viral infections delays virus clearance. In the present study, a potential role of macrophage activation in virus-induced SHL was supported by our results showing that corticosteroids inhibited virus-induced activation of macrophage functions in SCs and GERCs. Therefore, it is possible that this effect may be one of the pharmacological actions of corticosteroids in SHL. Our results also suggest that targeting macrophages or necroptosis may be effective for treatment of virus-induced SHL. This notion is supported by the results showing that macrophage depletion by Clopho or necroptosis inhibitors effectively suppressed HC death induced by virus infection. It is now considered that targeting macrophages is a potential therapy for many diseases including inflammatory diseases, metabolic diseases, and cancer [13,58]. Several methods have been developed to induce depletion, reprogramming, or repolarization of macrophages. Among them, nanotechnology-based systems (e.g., liposomes, dendrimers, gold nanoparticles, and polymeric nanoparticles) have been developed as specific delivery systems to the disease site [58]. Therefore, in patients with SHL, it will be important to develop a system that effectively delivers drugs to the inner ear through the blood-labyrinthine barrier. Moreover, it has been considered that necroptosis plays a crucial role in the regulation of various physiological processes and mediates various diseases such as ischemic brain injury, immunological disorders, and cancers. Therefore, development of therapeutic drugs targeting necroptosis has been carried out for various diseases [40,41]. In addition to these studies, our results suggest that necroptosis inhibitors may be effective therapeutic agents for virus-induced SSHL.

In conclusion, our results revealed novel TRAIL-mediated HC death induced by virus infection in cochlear sensory epithelium. Moreover, our results have shown that macrophage-targeting drugs and necroptosis inhibitors effectively protected HCs against virus infection in our ex vivo experimental system using cochlear sensory epithelia isolated from newborn mice. The weakness of our analysis is that it remains unclear whether this mechanism is also mediated by viruses that cause SHL other than TMEV, whether these treatments are also effective for SHL in mouse models, and the detailed induction mechanism of HC death. Moreover, it is unknown whether the same phenomenon seen in the inner ear of newborn mice applies to that of adult mice. However, we believe that our findings may shed new light on therapeutic paradigms for SSHL.

## Supporting information

**S1 Fig. Expression changes of genes correlated with HCs and SCs during development.**
Expression changes of genes correlated with HCs and SCs during development. (**A–D**) Shared

Harvard Inner-ear Laboratory Database (SHIELD; https://shield.hms.harvard.edu/index.html) is a resource for RNAseq datasets from HCs (GFP-positive cells) and their surrounding cells including SCs (GFP-negative cells) at E16, P0, P4, and P7. Here, we extracted *Trail* (**A**), *F4/80* (**B**), *Sox2* (**C**), and *Myo7a*, *prestin*, and *Pou4f3* (**D**) from this database. (**A**) *Trail* was expressed in SC fractions higher than HC fractions, especially at E16 and P0. (**B**) Macrophage marker *F4/80* was expressed in SC fractions. (**C**) SC marker *Sox2* was expressed in not only SC fractions, but also HC fractions during the embryonic stage, but the SC fractions maintained Sox2 expression, whereas the HC fractions did not after the postnatal stage. (**D**) HC markers *Myo7a*, *prestin*, and *Pou4f3* were expressed in HC fractions.
(TIF)

**S2 Fig. Gene Ontology analysis of microarray data showing upregulation of necroptosis- and apoptosis-related genes by TMEV infection.** Gene Ontology analysis of microarray data showing upregulation of necroptosis- and apoptosis-related genes by TMEV infection. We performed microarray analysis of mock cochlear sensory epithelia, LPS-treated cochlear sensory epithelia (9 and 16 h), and TMEV-infected cochlear sensory epithelia (9 and 16 h) and then examined necroptosis-, apoptosis- and ROS-related genes by Gene Ontology analysis. Among necroptosis-related genes, *Trail*, *Tlr3*, and *Mlkl* were upregulated in TMEV-infected cochlear sensory epithelia at 16 h compared with mock- and LPS-treated cochlear sensory epithelia. Apoptosis-related genes, such as *Stat1* and *Jun*, were upregulated in TMEV-infected cochlear sensory epithelia, especially at 16 h compared with mock- and LPS-treated cochlear sensory epithelia. However, ROS-related genes were not upregulated, except for *Nox2*, in TMEV-infected cochlear sensory epithelia compared with mock- and LPS-treated cochlear sensory epithelia.
(TIF)

## Acknowledgments

We thank H. Ozawa for helpful advice. We also thank Y. Abe, M. Shimizu, and C. Iwabuchi for discussions, and M. Kawagoe, Y. Asano, and T. Takatera for technical support.

## Author Contributions

**Conceptualization:** Yushi Hayashi, Nobuyuki Tanaka.

**Data curation:** Yushi Hayashi, Nobuyuki Tanaka.

**Formal analysis:** Yushi Hayashi, Nobuyuki Tanaka.

**Funding acquisition:** Yushi Hayashi, Nobuyuki Tanaka.

**Investigation:** Yushi Hayashi, Hidenori Suzuki, Wataru Nakajima, Ikuno Uehara, Atsuko Tanimura, Tatsuya Katsuno, Shin-ichiro Kitajiri, Naoto Koyanagi, Yasushi Kawaguchi, Koji Onomoto, Hiroki Kato, Mitsutoshi Yoneyama, Takashi Fujita, Nobuyuki Tanaka.

**Methodology:** Yushi Hayashi, Nobuyuki Tanaka.

**Resources:** Toshiki Himeda, Satoshi Koike, Mitsutoshi Yoneyama, Takashi Fujita, Nobuyuki Tanaka.

**Supervision:** Nobuyuki Tanaka.

**Validation:** Yushi Hayashi, Nobuyuki Tanaka.

**Visualization:** Yushi Hayashi, Nobuyuki Tanaka.

**Writing – original draft:** Yushi Hayashi, Nobuyuki Tanaka.

**Writing – review & editing:** Yushi Hayashi, Nobuyuki Tanaka.

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
