## [Decision Letter · Decision Letter 0]

29 Jun 2021

PONE-D-21-18184

Virus-infection in cochlear supporting cells induces audiosensory receptor hair cell death by TRAIL-induced necroptosis

PLOS ONE

Dear Dr. Tanaka,

Thank you for submitting your manuscript to PLOS ONE. After careful consideration, we feel that it has merit but does not fully meet PLOS ONE’s publication criteria as it currently stands. Therefore, we invite you to submit a revised version of the manuscript that addresses the points raised during the review process.

We look forward to receiving your revised manuscript.

Kind regards,

Alan Gi-Lun Cheng, M.D.

Academic Editor

PLOS ONE

Journal Requirements:

2. To comply with PLOS ONE submissions requirements, in your Methods section, please provide additional information on the animal research and ensure you have included details on (1) methods of sacrifice, (2) methods of anaesthesia and/or analgesia, and (3) efforts to alleviate suffering.

3. We note that you are reporting an analysis of a microarray, next-generation sequencing, or deep sequencing data set. PLOS requires that authors comply with field-specific standards for preparation, recording, and deposition of data in repositories appropriate to their field. Please upload these data to a stable, public repository (such as ArrayExpress, Gene Expression Omnibus (GEO), DNA Data Bank of Japan (DDBJ), NCBI GenBank, NCBI Sequence Read Archive, or EMBL Nucleotide Sequence Database (ENA)). In your revised cover letter, please provide the relevant accession numbers that may be used to access these data. For a full list of recommended repositories, see http://journals.plos.org/plosone/s/data-availability#loc-omics or http://journals.plos.org/plosone/s/data-availability#loc-sequencing.

7. PLOS requires an ORCID iD for the corresponding author in Editorial Manager on papers submitted after December 6th, 2016. Please ensure that you have an ORCID iD and that it is validated in Editorial Manager. To do this, go to ‘Update my Information’ (in the upper left-hand corner of the main menu), and click on the Fetch/Validate link next to the ORCID field. This will take you to the ORCID site and allow you to create a new iD or authenticate a pre-existing iD in Editorial Manager. Please see the following video for instructions on linking an ORCID iD to your Editorial Manager account: https://www.youtube.com/watch?v=_xcclfuvtxQ

Reviewers' comments:

Reviewer's Responses to Questions

**Comments to the Author**

1. Is the manuscript technically sound, and do the data support the conclusions?

Reviewer #1: Yes

Reviewer #2: Partly

Reviewer #3: Yes

2. Has the statistical analysis been performed appropriately and rigorously? 

Reviewer #1: Yes

Reviewer #2: Yes

Reviewer #3: Yes

3. Have the authors made all data underlying the findings in their manuscript fully available?

Reviewer #1: Yes

Reviewer #2: No

Reviewer #3: Yes

4. Is the manuscript presented in an intelligible fashion and written in standard English?

Reviewer #1: Yes

Reviewer #2: Yes

Reviewer #3: Yes

5. Review Comments to the Author

Reviewer #1: Sudden hearing loss is very common in clinic and Systemic corticosteroid administration is the primary treatment currently, although not all patients well respond to it, partly due to the poorly understood mechanisms underlying it. To obtain insights toward Sudden hearing loss, Hayashi et al in this study focused on characterizing the mechanisms underlying TMEV infection-caused hair cell death in cochlea. Briefly, each cochlear sample is incubated in medium for overnight before exposing to medium containing TMEV, which is an RNA virus. Three key findings are reported: 1) TRAIL expression in SCs is the key after TMEV infection; 2) Transformation of SCs (or GER/LER cells) into macrophages is another key step for HC death; 3) HC death occurs by necrosis, but not by apoptosis. The study is well-designed and clearly presented.

3 main comments are raised:

1) For the immunostaining images, the key single pannel is necessary. For instance, please show DR4 and DR5 single channel to better visualize their expression. In addition, a section view can be better demonstrate DR4 and 5’s expression in HCs. Furthermore, are DR4 and DR5 expressed in SCs?

2) How about the antibody specificity against TRAIL? Please provide evidence if possible. If the antibody is not specific, neutralizing experiments might have other explainations.

3) I also have a concern regarding whether TRAIL is critical in the process of HC death. In other words, as the authors speculate binding of TRAIL to DR4/5 is the key step, will HCs die in DR4 -/- or DR5 -/-, following TMEV infection? If answer is “ no”, more concerns are raised regarding the importance of binding of TRAIL with DR4 or DR5. The author should alert readers of this or provide further evidence to prove the direct involvement of TRAIL in TMEV induced HC death. Because it is one of the key findings in this study, it cannot be ignored.

Reviewer #2: The manuscript is interesting and presents data that suggest that certain viral infections may affect cochlear supporting cells in such a way that they respond via upregulation of Tnf-related apoptosis inducing ligand (TRAIL) which then binds to TRAIL receptors expressed by cochlear hair cells causing stereocilia degeneration and hair cell death. Some evidence is also presented to suggest that the hair cell death occurs via necroptosis and can be mitigated by treatment with anti-necroptosis agents or by inhibitors of macrophage activity. While the manuscript is largely novel and many of the findings appear sound, there are several aspects of the manuscript which need to be addressed before it can be made suitable for publication.

Specifically:

Major comments:

There are a lot of negative data, which may be fine, but the authors seem to interpret the lack of findings with more confidence than may be warranted without giving proper discussion to potential caveats such as insufficient doses, later timepoints, etc.

All of the hair cell counts have some issues. First, the TMEV treatment predominantly affects supporting cells which leads to disorganization. Such disorganization could lead to the appearance of lower numbers of hair cells when counted in windows that are 100um wide as the density of hair cells might be more severely affected than total hair cell number. Add this to the lack of cleaved caspase staining, and it becomes unclear the extent to which hair cells are dying. Though the MLKL staining counters this concern somewhat, all of these limitations should be discussed.

Second, the EM images suggest loss of bundles. If phalloidin was used to count surviving hair cells (as noted in the methods) then it is possible that loss of bundles would make the supposed loss of hair cells appear more exaggerated. It would be ideal if Myo7a counts could also be carried out and then there would not be ambiguity as a result of the stereocilia phenotype.

To the first point above, SC loss leading to disorganization, the authors have a note at one point about how the TRAIL neutralizing antibody treatment led to better hair cell survival even though SCs were still lost. However, there is no quantification of SCs in the paper. Actual quantification of the SCs would bolster this claim and help rebut the concern that disorganization is a main contributor to the observed phenotype rather than cell death.

Overall, much is left to be desired from the discussion. While the external information pertaining to necroptosis, macrophages, and TRAIL signaling is interesting, the manuscript would benefit greatly from added discussion that focuses specifically on the methodology and results of the studies therein, specifically consideration of what the strong conclusions are as well as any weaknesses, limitations, or other considerations. As noted above and below, this should include, but is not limited to, the choice of the virus used in this model, the methodology for counting of hair cells, and negative results.

No plan is mentioned for public access of the microarray data upon publication, rather the authors claim all data is in the manuscript, but given that a whole transcript GeneChip array was used, there should be a much larger dataset than what is included in supplemental figure 2. The data from this experiment should be uploaded to a publicly accessible database.

Minor comments:

The disorganization of bundles claimed to be elicited by recombinant TRAIL protein is not clear from the image (Fig 2c), single channel images of the phalloidin only should be shown, and ideally at higher magnification. Also, it is unclear whether the control is from a condition of 24 or 48 hours, but if only one control rather than 2 is to be presented, it should be from 48 hours.

Page 10, 2nd to last paragraph of the intro, it would be good to continue to make clear in this paragraph that the SCs are being likened to macrophages. Otherwise it is a bit unclear whether the authors are talking about possible roles for SCs or for actual macrophages which have been shown to reside in the inner ear and migrate to the sensory epithelium during times of stress or injury. While it becomes clear through the rest of the paper that the target is the SCs, making this more clear in this portion of the introduction will help limit confusion.

It would be preferable if the composition of the media (and possibly other details) of the explant procedure were outlined here rather than referring back to a reference from 2013.

TMEV (Theiler’s murine encephalitis virus) should be defined at its first use. Also, it could be helpful to justify the choice of this virus rather than murine CMV or others that might be more closely related to viruses known to cause hearing loss in humans.

p.11 line 86-87, what are the specified concentrations? These should be defined and justified, particularly for any that yielded negative results.

p. 15, TMEV is referred to as TEMV in multiple places. This persists through later portions of the manuscript. The abbreviation should be corrected to be consistent throughout.

p.15 line 188-192, wording is confusing, and should be framed more speculatively as these experiments are in vitro and hearing function was therefore not assessed. The authors can state that they speculate or hypothesize that TMEV infection might cause hearing loss prior to hair cell death since there is some evidence in vitro of bundle degeneration prior to hair cell death, but this would ultimately need to be validated in vivo.

Lines 198-211, this speculation about supporting cells producing TRAIL rather than hair cells should be moved to the discussion and presented as speculative with an acknowledgement that, in the absence of direct evidence, the authors do not know whether the increased TRAIL that was detected by qPCR is made by the SCs or not (no matter how reasonable such speculation may be).

Lines 234-235 this statement is an overreach… SCs could still be involved in hair cell death in response to aminoglycosides even if the specific transcripts that were examined in this study did not differ.

Lines 239- 240 should be ferroptosis and pyroptosis

In referencing the SHIELD data, the authors should be cautious in referring to “SC” expression as the dataset they are referring to only used a hair cell specific GFP, so the data they are referring to as “SC” gene expression is likely to include cell types other than just SCs, including macrophages or other immune cells.

Reviewer #3: This manuscript entitled virus-infection in cochlear supporting cells induces audiosensory receptor hair cell death by TRAIL-induced necroptosis, suggests a possible new target for preventing virus-induced sensorineural hearing loss. In this study, the authors show that supporting cells and GERCs induce hair cell death. This likely occurs via production of TRAIL as hair cell death is suppressed by TRAIL-neutralizing antibodies. Rather than through apoptotic mechanisms, this death occurs via necroptosis as it is inhibited by necroptosis inhibitors. Interestingly, corticosteroids also inhibited hair cell death via the inhibition of supporting cell/GERC transformation into macrophage-like cells. Hair cell death is also inhibited with macrophage depletion.

Overall, this is a well-written study. The authors do a nice job at systematically investigating the mechanism of hair cell death after viral infection. However, there are a few concerns that would need to be addressed prior to publication. Otherwise, this manuscript appears to make a significant contribution to the field.

In Figure 1F, the authors use TEM to show degeneration of hair cell stereocilia over time. Whereas the first panel shows stereocilia present at 16 hours, the second and third panels show loss of bundles by 21 hours prior to the loss of hair cells. They then claim that these data suggest hearing impairment is induced without hair cell death. However, the authors previously concluded in Figure 1A that hair cell death occurred at 16 hours post-viral infection, and that most hair cells died within 24 hours of infection. This leads me to think that these TEM scans may not be representative of the hair cell death process as proposed by the authors, especially if the numbers of hair cells at 24 hours as quantified in 1E are as low as ~1-2 IHC and 0 OHC in WT TMEV tissue.

In Figure 2, the authors report that hair cell death was suppressed by a TRAIL-neutralizing antibody. Whereas IHC numbers do not significantly differ between the Mock and the anti-TRAIL antibody, there is a statistically significant loss of OHC with the TRAIL-neutralizing antibody as compared to the Mock. This difference is not addressed by the authors.

The authors use gentamicin in Figure 3 as an ototoxic drug and analyze expression of markers in supporting cells in the presence of gentamicin. However, the figure legend only reports duration of gentamicin treatment and not dosage. The negative result of seeing no change in expression of these markers could certainly be due to a dose of gentamicin that is too low and/or incubation for too short a period of time. Further explanation and/or control experiments confirming appropriate ototoxic doses of gentamicin would be needed before supporting the conclusions made by the authors for this figure.

In Figure 4, the authors show that addition of necroptosis inhibitors necrostatin-1 and ponatinib suppress hair cell death. In Figure S2, they show that TEMV infection induces the expression of both apoptotic and necroptotic genes in the cochlear sensory epithelium. Expression of three necroptosis-related genes (Trail, Tlr3, Mlk1) increases in TMEV 16 hours, as does expression of numerous apoptosis-related genes. It would be interesting to note the changes to these genes in the presence of TMEV and necrostatin-1 or ponatinib. One would expect to see suppression of the necroptotic genes but not the apoptotic genes if these inhibitors had no effect on apoptosis.

The authors use dexamethasone in Figure 5 to suppress virus-induced hair cell death since steroids are the primary therapies for sudden sensorineural hearing loss. It would be interesting to see if prednisone, an alternative steroid used to treat SSHL with a much shorter half-life, has a similar effect. The authors note that dexamethasone inhibited hair cell damage, but not as strongly as necrostatin-1. Quantification of hair cell numbers comparing dexamethasone treatment versus necrostatin-1 treatment may be revealing to see whether this is truly the case. Since dexamethasone downregulates expression of macrophage markers in SCs and GERCs whereas necrostatin-1 inhibits necroptosis, it appears that the downstream signal—namely inhibition of necroptosis—contributes more significantly to the prevention of hair cell death than the upstream signal of suppressing macrophage expression in SCs and GERCs. Does this suggest that suppression of macrophage expression allows for activation of alternative mechanisms that still ultimately result in hair cell death via necroptosis? Answering these questions would be important to better elucidate whether targeting macrophages and/or necroptosis would be possible therapeutic avenues for the treatment of virus-induced SHL as described in lines 318-319.

Finally, a visual abstract or summary figure documenting the proposed mechanism of supporting cell-induced hair cell death would strongly enhance this paper.

6. PLOS authors have the option to publish the peer review history of their article (what does this mean?). If published, this will include your full peer review and any attached files.

Reviewer #1: No

Reviewer #2: No

Reviewer #3: No

---

## [Author Response · Author response to Decision Letter 0]

13 Oct 2021

Point-by-point responses to the reviewers’ comments

Ms No.: PONE-D-21-18184

Title: Virus infection in cochlear-supporting cells induces audiosensory receptor hair cell death by TRAIL-induced necroptosis

We are grateful for the invaluable comments and suggestions made by the referees. In accordance with their suggestions, we have added some new results to adequately address the raised issues and have amended the manuscript. Please find below our point-by-point responses to each of their comments.

Reviewer #1 

Sudden hearing loss is very common in clinic and Systemic corticosteroid administration is the primary treatment currently, although not all patients well respond to it, partly due to the poorly understood mechanisms underlying it. To obtain insights toward Sudden hearing loss, Hayashi et al in this study focused on characterizing the mechanisms underlying TMEV infection-caused hair cell death in cochlea. Briefly, each cochlear sample is incubated in medium for overnight before exposing to medium containing TMEV, which is an RNA virus. Three key findings are reported: 1) TRAIL expression in SCs is the key after TMEV infection; 2) Transformation of SCs (or GER/LER cells) into macrophages is another key step for HC death; 3) HC death occurs by necrosis, but not by apoptosis. The study is well-designed and clearly presented.

3 main comments are raised:

1) For the immunostaining images, the key single pannel is necessary. For instance, please show DR4 and DR5 single channel to better visualize their expression. In addition, a section view can be better demonstrate DR4 and 5’s expression in HCs. Furthermore, are DR4 and DR5 expressed in SCs?

In accordance with this comment, we have added a single channel image of DR4 or DR5 and a section view image of DR4 or DR5 in Fig. 2A. We have also shown the wide range of images that include SCs to present their expression in SCs clearer. In these images, significant expression of DR4 and DR5 was found in HCs, but rarely in SCs. Therefore, we have added the following sentence in the text: “Expression of DR4 and DR5 was found in HCs, but rarely in SCs (Fig. 2A)” (lines 209 to 210).

2) How about the antibody specificity against TRAIL? Please provide evidence if possible. If the antibody is not specific, neutralizing experiments might have other explainations.

In accordance with this comment, we have cited studies of the production of the N2B2 monoclonal antibody (Ref. 30) and its neutralization activity (Ref. 31), and added the following sentence in the text: “The specificity and neutralizing activity of the used antibody against TRAIL (monoclonal N2B2 antibody) have been shown previously [30, 31].” (lines 229 to 230).

3) I also have a concern regarding whether TRAIL is critical in the process of HC death. In other words, as the authors speculate binding of TRAIL to DR4/5 is the key step, will HCs die in DR4 -/- or DR5 -/-, following TMEV infection? If answer is “ no”, more concerns are raised regarding the importance of binding of TRAIL with DR4 or DR5. The author should alert readers of this or provide further evidence to prove the direct involvement of TRAIL in TMEV induced HC death. Because it is one of the key findings in this study, it cannot be ignored.

We agree with this comment. Indeed, we have not analyzed HC death in DR4-/- or DR5-/- after TMEV infection. Therefore, we revised the text as follows: “However, because four TRAIL receptors, which include DR4 and DR5, have been identified in humans [32], it is not to be elucidated whether only the binding of DR4 and DR5 expressed in HCs is important for HC death by TRAIL.” (lines 237 to 239).

Reviewer #2

The manuscript is interesting and presents data that suggest that certain viral infections may affect cochlear supporting cells in such a way that they respond via upregulation of Tnf-related apoptosis inducing ligand (TRAIL) which then binds to TRAIL receptors expressed by cochlear hair cells causing stereocilia degeneration and hair cell death. Some evidence is also presented to suggest that the hair cell death occurs via necroptosis and can be mitigated by treatment with anti-necroptosis agents or by inhibitors of macrophage activity. While the manuscript is largely novel and many of the findings appear sound, there are several aspects of the manuscript which need to be addressed before it can be made suitable for publication.

Specifically:

Major comments:

There are a lot of negative data, which may be fine, but the authors seem to interpret the lack of findings with more confidence than may be warranted without giving proper discussion to potential caveats such as insufficient doses, later timepoints, etc.

All of the hair cell counts have some issues. First, the TMEV treatment predominantly affects supporting cells which leads to disorganization. Such disorganization could lead to the appearance of lower numbers of hair cells when counted in windows that are 100um wide as the density of hair cells might be more severely affected than total hair cell number. Add this to the lack of cleaved caspase staining, and it becomes unclear the extent to which hair cells are dying. Though the MLKL staining counters this concern somewhat, all of these limitations should be discussed.

We agree with this comment. Therefore, in accordance with this comment, we have added the following sentences in the text: “We quantified HC death by the number of HCs along a longitudinal distance of 100 µm. However, TMEV treatment induced SC death, which resulted in disorganization of cochlear tissues. Therefore, it is possible that virus-induced disorganization led to the appearance of lower numbers of hair cells when counted in the same width. However, as shown in Fig. 4A, necrostatin treatment, especially 400 mm treatment, significantly inhibited loss of HC numbers (HC number was not markedly reduced compared with that observed without virus infection shown in Fig. 2B), whereas most SCs were lost. To more accurately assess HC death, we believe that the effects of disorganization in cochlear tissues need further analysis.” (lines 279 to 287).

Second, the EM images suggest loss of bundles. If phalloidin was used to count surviving hair cells (as noted in the methods) then it is possible that loss of bundles would make the supposed loss of hair cells appear more exaggerated. It would be ideal if Myo7a counts could also be carried out and then there would not be ambiguity as a result of the stereocilia phenotype.

To the first point above, SC loss leading to disorganization, the authors have a note at one point about how the TRAIL neutralizing antibody treatment led to better hair cell survival even though SCs were still lost. However, there is no quantification of SCs in the paper. Actual quantification of the SCs would bolster this claim and help rebut the concern that disorganization is a main contributor to the observed phenotype rather than cell death.

Owing to our mistake, the initial Fig. 2B was confusing because it only showed HCs. As presented in our revised Fig. 2B that shows a wider range, almost all SCs were lost after virus infection for 24 hours as determined by loss of DAPI-stained cells. However, HCs were still observed in the presence of the TRAIL-neutralizing antibody. We believe that the revised Fig. 2B will help readers to understand this phenomenon.

Overall, much is left to be desired from the discussion. While the external information pertaining to necroptosis, macrophages, and TRAIL signaling is interesting, the manuscript would benefit greatly from added discussion that focuses specifically on the methodology and results of the studies therein, specifically consideration of what the strong conclusions are as well as any weaknesses, limitations, or other considerations. As noted above and below, this should include, but is not limited to, the choice of the virus used in this model, the methodology for counting of hair cells, and negative results.

In accordance with this comment, we have added the following description in the Discussion: “In conclusion, our results revealed novel TRAIL-mediated HC death induced by virus infection in cochlear sensory epithelium. Moreover, our results have shown that macrophage-targeting drugs and necroptosis inhibitors effectively protected HCs against virus infection in our ex vivo experimental system using cochlear sensory epithelia isolated from newborn mice. The weakness of our analysis is that it remains unclear whether this mechanism is also mediated by viruses that cause SHL other than TMEV, whether these treatments are also effective for SHL in mouse models, and the detailed induction mechanism of HC death. Moreover, it is unknown whether the same phenomenon seen in the inner ear of newborn mice applies to that of adult mice. However, we believe that our findings may shed new light on therapeutic paradigms for SSHL.” (lines 372 to 380).

No plan is mentioned for public access of the microarray data upon publication, rather the authors claim all data is in the manuscript, but given that a whole transcript GeneChip array was used, there should be a much larger dataset than what is included in supplemental figure 2. The data from this experiment should be uploaded to a publicly accessible database.

In accordance with this comment, we have added the following description to the text: “These data have been uploaded to the Gene Expression Omnibus (GEO) database (https://www.ncbi.nlm.nih.gov/geo/) under accession code GEO: GSE89556” (lines 157 to 159).

Minor comments:

The disorganization of bundles claimed to be elicited by recombinant TRAIL protein is not clear from the image (Fig 2c), single channel images of the phalloidin only should be shown, and ideally at higher magnification. Also, it is unclear whether the control is from a condition of 24 or 48 hours, but if only one control rather than 2 is to be presented, it should be from 48 hours.

In accordance with this comment, we have added a single channel image of phalloidin for each condition. The control is from the condition of 48 hours. We have added this information in Fig. 2C.

Page 10, 2nd to last paragraph of the intro, it would be good to continue to make clear in this paragraph that the SCs are being likened to macrophages. Otherwise it is a bit unclear whether the authors are talking about possible roles for SCs or for actual macrophages which have been shown to reside in the inner ear and migrate to the sensory epithelium during times of stress or injury. While it becomes clear through the rest of the paper that the target is the SCs, making this more clear in this portion of the introduction will help limit confusion.

In accordance with this comment, we have added the following description to the second to last paragraph of the Introduction: “In relation to these findings, we found that cochlear SCs function similarly to tissue resident macrophages that protect HCs from pathogens [12].” (lines 59 to 60).

It would be preferable if the composition of the media (and possibly other details) of the explant procedure were outlined here rather than referring back to a reference from 2013.

We prepared Dulbecco’s modified Eagle’s medium (DMEM; Sigma-Aldrich Inc., St. Louis, MO, USA) supplemented with D-glucose (6 g/l) and penicillin G (150 �g/ml) for explant cultures. We have added this information to the Materials and Methods (lines 82 to 83).

TMEV (Theiler’s murine encephalitis virus) should be defined at its first use. Also, it could be helpful to justify the choice of this virus rather than murine CMV or others that might be more closely related to viruses known to cause hearing loss in humans.

In accordance with this comment, we have added the following description to the introduction of TMEV: “We previously found that Theiler’s murine encephalomyelitis virus (TMEV) infection in isolated murine newborn cochlear sensory epithelium induces IFN-α/β production [11]. TMEV is a small RNA picornavirus commonly used as an experimental model system for blood-brain barrier disruption [20]. Recently, we observed that TMEV infection is mainly established in SCs and HC infection is rarely observed in the presence of IFN-α/β produced by SCs that function as macrophage-like cells [12]. We also observed that SC infection is established in the early stage of TMEV infection (9 h after virus infection) and GERC infection is established in the later stage (16 h after virus infection) [12]. To understand the influence of virus infection in SCs on HCs, we analyzed the cell status of HCs using the same experimental system. It has been reported that TMEV infects macrophages [21]. Indeed, we have observed that TMEV infected almost all SCs and GERCs in our experimental system [12]. Therefore, although TMEV is not a virus that causes SHL in humans such as CMV, we used this experimental system to investigate the effects of infection with cochlear SCs that are protective cells against virus infection in mice.” (underlined text is newly written; lines 174 to 187).

p.11 line 86-87, what are the specified concentrations? These should be defined and justified, particularly for any that yielded negative results.

The concentration of the TRAIL-neutralizing antibody was 0.01 mg/ml. We have added this information to the Materials and Methods (line 90).

p. 15, TMEV is referred to as TEMV in multiple places. This persists through later portions of the manuscript. The abbreviation should be corrected to be consistent throughout.

This was a mistake. In accordance with this comment, we have carefully checked and rewritten the text.

p.15 line 188-192, wording is confusing, and should be framed more speculatively as these experiments are in vitro and hearing function was therefore not assessed. The authors can state that they speculate or hypothesize that TMEV infection might cause hearing loss prior to hair cell death since there is some evidence in vitro of bundle degeneration prior to hair cell death, but this would ultimately need to be validated in vivo.

We agree with this comment. Therefore, we have deleted the sentence “which suggested that this HC death signal induced hearing impairment without HC death” from the text.

Lines 198-211, this speculation about supporting cells producing TRAIL rather than hair cells should be moved to the discussion and presented as speculative with an acknowledgement that, in the absence of direct evidence, the authors do not know whether the increased TRAIL that was detected by qPCR is made by the SCs or not (no matter how reasonable such speculation may be).

In accordance with this comment, we have added the following description to the text: “These findings suggest that TRAIL was produced by SCs, which function as macrophages, after TEMV infection. However, it cannot be ruled out that factor(s) produced by virus-infected SCs act on HCs to induce TRAIL.” (lines 225 to 227).

Lines 234-235 this statement is an overreach… SCs could still be involved in hair cell death in response to aminoglycosides even if the specific transcripts that were examined in this study did not differ.

We agree with this comment and have changed the following description from “These results suggest that virus-induced HC death differs from that induced by ototoxic drugs, and SCs and GERCs are not involved in aminoglycoside-induced HC death” to “These results suggest that the virus-induced HC death mechanism differs from that induced by ototoxic drugs” (lines 258 to 259).

Lines 239- 240 should be ferroptosis and pyroptosis

This was a mistake and we have carefully checked and rewritten the text.

In referencing the SHIELD data, the authors should be cautious in referring to “SC” expression as the dataset they are referring to only used a hair cell specific GFP, so the data they are referring to as “SC” gene expression is likely to include cell types other than just SCs, including macrophages or other immune cells.

In accordance with this comment, we have added the following description to the text: “The SC fraction is a cell population other than the HC fraction of sensory epithelial cells, which was thought to be absent of immune cells, but it cannot be completely ruled out this population contains small numbers of macrophages and lymphocytes. However, these findings suggest that TRAIL was produced by SCs that function as macrophages after TEMV infection.” (lines 221 to 225).

Reviewer #3

This manuscript entitled virus-infection in cochlear supporting cells induces audiosensory receptor hair cell death by TRAIL-induced necroptosis, suggests a possible new target for preventing virus-induced sensorineural hearing loss. In this study, the authors show that supporting cells and GERCs induce hair cell death. This likely occurs via production of TRAIL as hair cell death is suppressed by TRAIL-neutralizing antibodies. Rather than through apoptotic mechanisms, this death occurs via necroptosis as it is inhibited by necroptosis inhibitors. Interestingly, corticosteroids also inhibited hair cell death via the inhibition of supporting cell/GERC transformation into macrophage-like cells. Hair cell death is also inhibited with macrophage depletion.

Overall, this is a well-written study. The authors do a nice job at systematically investigating the mechanism of hair cell death after viral infection. However, there are a few concerns that would need to be addressed prior to publication. Otherwise, this manuscript appears to make a significant contribution to the field.

In Figure 1F, the authors use TEM to show degeneration of hair cell stereocilia over time. Whereas the first panel shows stereocilia present at 16 hours, the second and third panels show loss of bundles by 21 hours prior to the loss of hair cells. They then claim that these data suggest hearing impairment is induced without hair cell death. However, the authors previously concluded in Figure 1A that hair cell death occurred at 16 hours post-viral infection, and that most hair cells died within 24 hours of infection. This leads me to think that these TEM scans may not be representative of the hair cell death process as proposed by the authors, especially if the numbers of hair cells at 24 hours as quantified in 1E are as low as ~1-2 IHC and 0 OHC in WT TMEV tissue.

As the reviewer commented, our study did not provide sufficient results to consider the association between loss of stereocilia and HC death. Therefore, we have deleted the following comment in the text: “Interestingly, prior to cell death, there was loss of the critical structures involved in sound signal transduction, namely sensory hairs (Fig. 1F), which suggested that this HC death signal induced hearing impairment without HC death”.

In Figure 2, the authors report that hair cell death was suppressed by a TRAIL-neutralizing antibody. Whereas IHC numbers do not significantly differ between the Mock and the anti-TRAIL antibody, there is a statistically significant loss of OHC with the TRAIL-neutralizing antibody as compared to the Mock. This difference is not addressed by the authors.

In accordance with this comment, we have added the following description to the text: “In this result, the decrease of IHCs was almost suppressed by the TRAIL-neutralizing antibody, but a decrease of OHCs was slightly observed (Fig. 2B). Regarding this difference, it is possible that there was a difference in the local TRAIL concentration and the effect of SC loss, but this has not been clarified at this time.” (lines 232 to 236).

The authors use gentamicin in Figure 3 as an ototoxic drug and analyze expression of markers in supporting cells in the presence of gentamicin. However, the figure legend only reports duration of gentamicin treatment and not dosage. The negative result of seeing no change in expression of these markers could certainly be due to a dose of gentamicin that is too low and/or incubation for too short a period of time. Further explanation and/or control experiments confirming appropriate ototoxic doses of gentamicin would be needed before supporting the conclusions made by the authors for this figure.

Tao et al. mentioned in their study that cochlear sensory epithelia were treated with 0.5 mM gentamicin for 3 h and then gentamicin was washed out and replaced with fresh medium. While no detectable hair cell loss was observed at 3 h, severe hair cell damage was caused by gentamicin at 24 h. We believe that these results demonstrate the validity of this model of explant culture to compare with that of virus infection.

In Figure 4, the authors show that addition of necroptosis inhibitors necrostatin-1 and ponatinib suppress hair cell death. In Figure S2, they show that TEMV infection induces the expression of both apoptotic and necroptotic genes in the cochlear sensory epithelium. Expression of three necroptosis-related genes (Trail, Tlr3, Mlk1) increases in TMEV 16 hours, as does expression of numerous apoptosis-related genes. It would be interesting to note the changes to these genes in the presence of TMEV and necrostatin-1 or ponatinib. One would expect to see suppression of the necroptotic genes but not the apoptotic genes if these inhibitors had no effect on apoptosis.

To show the effect of the necroptosis inhibitor on SC death, we changed Fig. 4A to a wide range of images that included SCs. As shown in this revised figure, although necrostatin-1 effectively suppressed HC loss, there was no suppression of SC loss. This result suggested that the necroptosis inhibitor affected HC death, but did not inhibit SC death, which was caused by apoptosis (Fig. 3A). We have not investigated the effect of the necroptosis inhibitor on the expression of apoptosis-related genes, but believe that this result shows that it does not affect induction of apoptosis. We have added the following comment to the text: “Moreover, although necrostatin-1 effectively suppressed HC loss, there was no suppression of SC loss (Fig. 4A). This result suggests that the necroptosis inhibitor affected HC death, but did not inhibit SC death, which was caused by apoptosis (Fig. 3A).” (lines 268 to 271).

The authors use dexamethasone in Figure 5 to suppress virus-induced hair cell death since steroids are the primary therapies for sudden sensorineural hearing loss. It would be interesting to see if prednisone, an alternative steroid used to treat SSHL with a much shorter half-life, has a similar effect. The authors note that dexamethasone inhibited hair cell damage, but not as strongly as necrostatin-1. Quantification of hair cell numbers comparing dexamethasone treatment versus necrostatin-1 treatment may be revealing to see whether this is truly the case. Since dexamethasone downregulates expression of macrophage markers in SCs and GERCs whereas necrostatin-1 inhibits necroptosis, it appears that the downstream signal—namely inhibition of necroptosis—contributes more significantly to the prevention of hair cell death than the upstream signal of suppressing macrophage expression in SCs and GERCs. Does this suggest that suppression of macrophage expression allows for activation of alternative mechanisms that still ultimately result in hair cell death via necroptosis? Answering these questions would be important to better elucidate whether targeting macrophages and/or necroptosis would be possible therapeutic avenues for the treatment of virus-induced SHL as described in lines 318-319.

This is an error in our description. In fact, the effectiveness of necroststin-1 and dexamethasone cannot be compared in this experiment. Indeed, there are no results to deny that suppression of macrophages allows activation of alternative mechanisms that still ultimately result in hair cell death. We have described this in the Discussion (third paragraph) that both macrophage-targeting therapy and a necroptosis inhibitor may be candidates for SHL treatment without superiority and we believe that these statements will not mislead readers. We have changed the description in the Figure 5 legend from “Corticosteroid dexamethasone (Dex) also inhibited HC damage, but not as strongly as necrostatin-1 (Nec)” to “Corticosteroid dexamethasone (Dex) inhibited HC damage”.

Finally, a visual abstract or summary figure documenting the proposed mechanism of supporting cell-induced hair cell death would strongly enhance this paper.

In accordance with this comment, we have added a visual abstract.

---

## [Decision Letter · Decision Letter 1]

10 Nov 2021

Virus-infection in cochlear supporting cells induces audiosensory receptor hair cell death by TRAIL-induced necroptosis

PONE-D-21-18184R1

Dear Dr. Tanaka,

We’re pleased to inform you that your manuscript has been judged scientifically suitable for publication and will be formally accepted for publication once it meets all outstanding technical requirements.

Kind regards,

Alan G. Cheng, M.D.

Academic Editor

PLOS ONE

Additional Editor Comments (optional):

Reviewers' comments:

Reviewer's Responses to Questions

**Comments to the Author**

1. If the authors have adequately addressed your comments raised in a previous round of review and you feel that this manuscript is now acceptable for publication, you may indicate that here to bypass the “Comments to the Author” section, enter your conflict of interest statement in the “Confidential to Editor” section, and submit your "Accept" recommendation.

Reviewer #1: All comments have been addressed

Reviewer #3: All comments have been addressed

2. Is the manuscript technically sound, and do the data support the conclusions?

Reviewer #1: Yes

Reviewer #3: Yes

3. Has the statistical analysis been performed appropriately and rigorously? 

Reviewer #1: Yes

Reviewer #3: Yes

4. Have the authors made all data underlying the findings in their manuscript fully available?

Reviewer #1: Yes

Reviewer #3: Yes

5. Is the manuscript presented in an intelligible fashion and written in standard English?

Reviewer #1: Yes

Reviewer #3: Yes

6. Review Comments to the Author

Reviewer #1: My concerns are addressed in a satisfied way. It is an important topic in Sudden hearing loss filed, but whether the DR4 and DR5 (or other receptors) are indeed critical for HC death remains unclear in vivo, because solid genetic evidence are missed in this study. But I am ok to publish this initial study, and more thorough studies could be followed.

Reviewer #3: (No Response)

7. PLOS authors have the option to publish the peer review history of their article (what does this mean?). If published, this will include your full peer review and any attached files.

Reviewer #1: No

Reviewer #3: No

---

## [Editor Report · Acceptance letter]

16 Nov 2021

PONE-D-21-18184R1 

Virus-infection in cochlear supporting cells induces audiosensory receptor hair cell death by TRAIL-induced necroptosis 

Dear Dr. Tanaka:

I'm pleased to inform you that your manuscript has been deemed suitable for publication in PLOS ONE. Congratulations! Your manuscript is now with our production department. 

Kind regards, 

on behalf of

Dr. Alan G. Cheng 

Academic Editor

PLOS ONE